# Lossy Mode Resonance Sensors Fabricated by RF Magnetron Sputtering GZO Thin Film and D-Shaped Fibers

**Chuen-Lin Tien** [1,2,*] , **Tzu-Chi Mao** [2] **and Chi-Yuan Li** [1]

[1] Department of Electrical Engineering, Feng Chia University, Taichung 40724, Taiwan; fishball2437@gmail.com
[2] Program of Electrical and Communications Engineering, Feng Chia University, Taichung 40724, Taiwan; sightlyopti@gmail.com
* Correspondence: cltien@fcu.edu.tw; Tel.: +886-4-24517250 (ext. 3809)

**Abstract:** We demonstrate a new refractive index (RI) and salinity sensor based on a lossy mode resonance (LMR) effect which combines fiber-optic side-polishing and radio-frequency (RF) sputtering techniques. The side-polished fiber can enhance optical fibers to generate an evanescent field in sensing applications. Gallium-doped zinc oxide (GZO) thin films produce a high attenuation lossy mode resonance effect that permits a highly sensitive refractive index and salinity fiber sensor. GZO thin film was prepared by an RF magnetron sputtering method. The thickness of the D-shaped fiber sensing device was 74.7 μm, and a GZO film thickness of 67 nm was deposited on the polished surface of the D-shaped fiber to fabricate LMR type liquid salinity sensors. The sensitivity of 3637.8 nm/RIU was achieved in the RI range of 1.333 to 1.392. To investigate the sensitivities of LMR salinity sensors, the NaCl solution salinities of 0%, 50%, 100%, 150%, 200%, and 250% were measured in this work. The experimental result shows that the sensitivity of the salinity sensor is 0.964 nm per salinity unit (SU).

**Keywords:** refractive index sensor; salinity; side-polished fiber; gallium-doped zinc oxide; lossy mode resonance

## 1. Introduction

Detection of the liquid refractive index (RI) is an important issue in chemical, biological, and environmental engineering fields. Optical fiber sensors have demonstrated numerous features, for example small size, light-weight, fast response, and anti-electromagnetic interference [1,2]. There are many approaches used to measure the liquid refractive index, such as fiber Bragg grating [3], surface plasmon resonance (SPR) [4], interferometric technology [5], etc. However, the sensitivity and dynamic detection range of the refractive index measuring methods are limited [6]. Thus, we propose a cost-effective approach for sensing changes in refractive indices by using lossy mode resonance (LMR) effect to overcome this limitation.

Salinity is a key parameter to determine the density of seawater in determining many aspects of the chemistry of waters and soils. In a theoretical aspect, the density and absolute salinity can be evaluated by a direct measurement of the refractive index, but the absolute salinity of seawater is difficult to measure directly. The most common way to evaluate salinity is to measure the amount of salt in 1000g of water, so that the salinity is referred to as parts per thousand (%) or ppt (1 ppt = 1000 mg/L). Most oceans have salinity between 34% and 36% [7]. Several methods have been reported for salinity measurement. In 1995, Min et al. proposed an ultrasonic technique [8] based on the measurement of the travel time of light to measure the salinity. In 1999, Diniz et al. [9] reported a polyaniline matrix

coated wire electrode for salinity measurement in a range from 0.010% to 75%. Esteban et al. [10] reported an RI optical sensor combined with a surface plasmon excitation to measure the salinity of water. Rahman [11] demonstrated a simple intensity modulated displacement sensor to detect the salinity of sodium chloride solutions varying from 0 to 12% by using a beam-through technique. Fiber optic salinity sensors based on salinity sensitive hydrogel-coated fiber Bragg gratings (FBG) [12] have been reported. Men et al. [13] proposed a fiber-optic sensing system consisting of FBGs coated with different polymers to monitor the salinity. However, these approaches cannot achieve a cost-effective and compact structure to fabricate easily. Optical fiber sensors have been used extensively in different engineering fields due to their many desirable advantages [14]. Compared to traditional sensors these have better sensitivity, accuracy, and reliability. The fiber-based sensors possess the potential of high sensitivity, such as the side-polished fiber sensor [15], thin-film type fiber sensor [16], Bragg grating fiber sensor [17], as well as the long-period grating fiber sensor [18].

The light propagation in the ordinary optical fibers cannot interact with the fiber's surrounding medium. In order to achieve this interaction, one must generate the evanescent field by different side-polished fiber configurations. As the cladding of a fiber is removed to within a few microns, the evanescent field can interact with the ambient refractive indices. Therefore, the D-shaped fibers can be used to enhance the evanescent field magnitude and sensing sensitivity. Thin film was coated on the D-shaped fibers to make a resonance-based fiber sensor. In general, if a thin film deposited on the side-polished surface of optical fiber devices will generate two kinds of resonances, one is surface plasmon resonance (SPR) [19]; the other is lossy mode resonance (LMR). The well-known resonance phenomenon caused by these modes is SPR sensors. However, there are other types of modes supported by absorbing thin films. Yang and Sambles [20] called them guided modes. They reported that a thin film supports the LMR effect if the real part of its permittivity is positive and higher in magnitude than both its own imaginary part and the material surrounding the thin film. The evanescent field of a propagation wave in a D-shaped fiber is accessible through a removed part of the cladding. LMR is considered as the standing electromagnetic wave confined between the two surfaces of the waveguide [21]. For example, transparent conductive oxides (TCO) are good candidates for supporting LMR thanks to the combination of conductive and transparent properties in the visible/infrared region. Gallium-doped ZnO (GZO) films, with low resistivity and high transmittance in the visible and near infrared spectrums, have been prepared by different techniques [22–28]. For example, microwave assisted growth (MAG) was used to fabricate gallium-doped ZnO nanostructure and to tailor optoelectronic characteristics [25]. Rana et al. demonstrated gallium-doped ZnO nanorods optimized by controlling $OH^-$ ion supplying to the solution via $NH_4OH$ decomposition [26]. Ko et al. reported the structural and optical properties of gallium-doped ZnO films grown on GaN templates by plasma-assisted molecular-beam epitaxy [27]. Park et al. investigated the properties of GZO thin films by a pulsed laser deposition method [28]. Among the various techniques, radio-frequency magnetron sputtering offers dense, uniform, and well-adhered films. It has controllable parameters and the most effective processes for the deposition of high quality thin-film materials. In this work, we chose GZO thin film with a high refractive index as a sensing material that produces the LMR phenomenon.

Due to the liquid refractive index changes being proportional to its density, which is strongly correlated with salinity, the measurement of seawater's refractive index can be used to measure its salinity. Little research has been done on the measurement of salinity concentration and the refractive index of salt water. Therefore, we proposed a cost-effective fiber-optic sensor based on the side-polished fiber as well as GZO thin-film coatings. We also investigated GZO thin films deposited on the D-shaped fibers with different thicknesses to compare the sensitivity of LMR sensors. The influence of the GZO film thickness on the LMR sensor's sensitivity was presented. The RI fiber-optic sensor with high sensitivity can be achieved.

The aim of this work is to investigate the gallium-doped zinc oxide as LMR-supporting thin film material. Zinc oxide thin films have already been widely used for fabrication of various sensors [29]. If the refractive index of GZO thin film layer is sensitive to the surrounding medium, a wavelength

shift can be observed by the LMR effect. The proposed LMR type fiber-optic sensor demonstrates a high performance in the RI and salinity measurement. Therefore, an LMR-based RI sensor is helpful for the fabrication of several kinds of sensors.

## 2. Materials and Methods

### 2.1. Preparation of Gallium-Doped Zinc Oxide Thin Film

Gallium-doped zinc oxide (GZO) thin films were prepared by radio-frequency (RF) magnetron sputtering technique. A $Ga_2O_3$: ZnO target with a purity of 99.99% was used as the coating material. D-shaped fibers, BK7 glass, and silicon wafer were used as coating substrates. The distance between the substrates and the target was 95 mm. The GZO target with a diameter of 50.8 mm and a thickness of 3 mm were powered by a radio frequency supply. During thin film deposition, the sputtering power of the ZnO target was kept at 80 W. Pure argon gas with a flow rate of 20 sccm was used as the sputtering gas, and the working pressure was fixed at 0.27 Pa. The film thickness was measured using a surface profilometer (Surfcoder, ET-3000, Kosaka Laboratory Ltd., Tokyo, Japan). The optical device quality can be evaluated by the optical transmittance. The optical transmittance was measured using a UV–VIS–NIR spectrophotometer (Shimadzu UV-3100 PC, Shimadzu Co. Ltd., Kyoto, Japan) in the wavelength range from 300 nm to 2500 nm. The films' structure was examined by X-ray diffractometry (XRD, Bruker D8 Discover, Bruker Co., Karlsruhe, BW, Germany). The microstructure was analyzed by a high-resolution scanning electron microscopy (SEM, Hitachi S3000, Hitachi Co. Ltd., Tokyo, Japan).

### 2.2. Fabrication of LMR Fiber-Optic Sensors

The choice of thin film materials and coating technology is an essential factor that influences the fiber sensor performance. When a thin-film deposited on the side-polished fibers, it can be regarded as a waveguide structure and affects the propagation light. A common surface plasmon resonance (SPR) arises if the real part of the thin-film permittivity is negative and higher in magnitude than its own imaginary part and the permittivity of the surrounding medium [30]. In this case, coupling appears between light propagating through the wave-guide and a surface plasmon. The SPR can only use a transverse-magnetic (TM) polarized light source. On the other hand, the LMR phenomenon arises as the real part of the thin-film permittivity is positive and higher in magnitude than both its own imaginary part and the permittivity of the medium surrounding the thin film. One of the characteristics show that the LMR effect can be generated by either transverse electric (TE) or TM polarized light. Furthermore, multiple resonances are generated without modifying the optical fiber structure, and they can be obtained with a wide variety of materials. Villar et al. [31] reported that the propagation light in semiconductor cladding waveguides possesses some attenuation maxima for specific thickness values of the semiconductor cladding and at certain wavelengths of incidence values. This reason is a coupling between the waveguide modes and a specific lossy mode of the semiconductor layer [32].

Thin films coated onto D-shaped fibers to meet the resonance conditions can generate an LMR effect [33]. LMR can be generated by metal oxides, such as indium tin oxide (ITO) [34], $TiO_2$ [35], and indium oxide [36]. Since these thin films are characterized by their relative permittivity, thin films absorb light and are thus characterized by a complex value of relative permittivity. The dielectric constant of GZO thin films is defined by the Drude–Lorentz model [37] and can be expressed as:

$$\varepsilon(\omega) = \varepsilon_\infty - \frac{\omega_p^2}{\omega^2 + i(\omega/\tau)} + \frac{s_0\omega_0^2}{\omega_0^2 - \omega^2 - i\gamma\omega},$$ (1)

where $\varepsilon_\infty$ is the high frequency dielectric constant, and the literature value of $\varepsilon_\infty$ is 3.5–3.7 [38]. $\omega$ is an angular frequency of a time-dependent electric field and $\omega_p$ is the plasma frequency. $\tau$ is the electronic scattering time. $s_0$ is the oscillator strength, $\omega_0$ is the oscillator resonance frequency, and

$\gamma$ is the oscillator damping constant. Regarding the complex dielectric constant, a real part and an imaginary part can be expressed as:

$$\varepsilon_R = \varepsilon_\infty - \frac{\omega_p^2 \omega^2}{\omega^4 + (\omega/\tau)^2} + \frac{s_0 \omega_0^4 - s_0 \omega^2 \omega_0^2}{\omega_0^4 - 2\omega^2 \omega_0^2 + \omega^4 + \gamma^2 \omega^2} \tag{2}$$

and

$$\varepsilon_I = -\frac{\omega_p^2 (\omega/\tau)}{\omega^4 + (\omega/\tau)^2} i + \frac{s_0 \omega_0^2 \gamma \omega}{\omega_0^4 - 2\omega^2 \omega_0^2 + \omega^4 + \gamma^2 \omega^2} i. \tag{3}$$

If GZO thin films are coated on the side-polished D-shaped fiber, then its real part of the refractive index is higher than that of the D-shaped fiber in the wavelength region. In addition, the index of the surrounding medium (salt water) above the GZO film is ~1.333. In this work, we proposed fiber-optic sensing structure to produce an LMR effect. The guided modes supported by the proposed sensing structure, and the resonance modes increase as the GZO film's thickness increases. However, the loss of guided mode reaches the maximum as the GZO film thickness is corresponding to the cut-off thickness for a specific mode [39]. If thin films were coated onto a single mode fiber (SMF), then the lossy modes for either TE or TM reached a maximum at its particular thickness [40]. In this case, coupling occurs between the guided mode and the lossy mode in the absorbing films. In other words, the phase matching between the guided mode supported by a D-shaped fiber and the lossy mode supported by the GZO thin film is achieved.

## 3. Results and Discussion

Two spectrometers (Ocean Optics NIR 512 and USB 4000, Ocean Optics Inc., Largo, FL, USA) were utilized to study both the visible light and the near infrared spectrum regions. The goal of high sensitivity can be reached by thinning the single mode fibers (SMF) to produce an evanescent wave to interact with the surrounding medium within the penetration depth. The D-shaped fibers must be polished precisely to form a smooth surface and to deposit GZO thin film. The core is very close to the flat surface of the D-shaped optical fibers. If a D-shaped fiber is polished to be thin enough, the evanescent wave can interact with an analyte of the surroundings. The side-polishing section of D-shaped fiber is sensitive to the refractive index of surrounding medium. In the fabrication of side-polishing D-shaped fibers with different remaining thicknesses (approximately 70–75 μm), single mode fibers (SMF-130 v) were polished using a homemade fiber-polishing system. The SMF cladding was removed until the strong evanescent wave was produced. The surface roughness of the side-polishing D-shaped fibers have to reduce to avoid scattering loss. After finishing optical fiber side-polishing, the remaining thickness of the D-shaped fibers was measured by using a high magnification optical image microscope. The remaining thickness of the D-shaped fibers was about 74.7 μm. The polished fiber length of 30 mm was made for the sensing device. The schematic of our proposed LMR sensing device based on D-shaped fiber is illustrated in Figure 1.

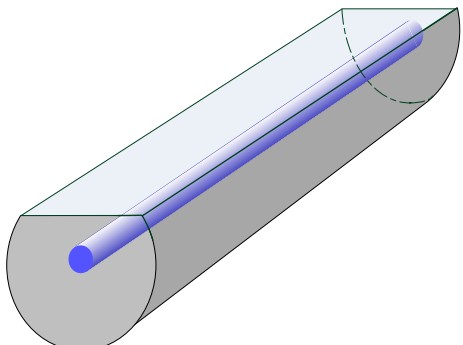

**Figure 1.** The proposed LMR sensing device based on a D-shaped fiber.

Before GZO coatings, the silicon wafer, glass substrates, and D-shaped fibers were ultrasonically cleaned. The D-shaped fibers were embedded and glued onto a coating substrate. Then, GZO thin film was coated onto the side-polishing fiber surface by using an RF magnetron sputtering method. The thickness of GZO film was 67 nm which was measured by a surface profilometer. The microstructure was analyzed by a scanning electron microscopy (SEM). Figure 2 shows the SEM micro-image of the D-shaped fiber coated with GZO thin film. In the present experiment, distilled water was used to avoid any type of contamination error. Similarly, the measured solutions that had different concentrations of sodium chloride (NaCl) were prepared. Two kinds of experiments were conducted in dilute NaCl solutions. The first kind was to detect the refractive indices (RI) of NaCl solutions from RI = 1.333 to 1.392. The RI of NaCl solutions was determined by an Abbe refractometer (DR-A1-Plus, ATAGO Co. Ltd., Kyoto, Japan) with a resolution of $1 \times 10^{-4}$ for indices of 1.33 to 1.58. The second experiment was the salinity measurement. The proposed sensor was immersed in different concentrations of NaCl solutions with a range of salinities from 0% to 250%. This measuring salinity range is larger than the previously published papers for salinity measurements. The above fiber-optic sensing region was immersed into sensing solutions and the transmission spectrum date was analyzed by a computer program, respectively. The variation in the optical spectrum of the GZO film deposited on the side-polished fibers with different salinity solutions was measured. Figure 3 shows the schematic diagram of the experimental setup with a halogen light source, spectrometers, and a D-shaped fiber coated with GZO thin film. A white light source was linked to an LMR fiber-optic sensor with the other end attached to two spectrometers (both from Ocean Optics, USB 4000 and NIR 512). The output spectrum of the spectrometer was observed in the wavelength range of 350 to 1700 nm. In order to observe the wavelength shift of the LMR absorption peaks, the side-polishing section (about 30 mm) is immersed in different NaCl solutions. For salinity detection experiments, the sensitive region of the sensors was immersed into saline solutions at different salinity indices (0, 50, 100, 150, 200 and 250%). We changed the sensing solutions and analyzed the data measured by the optical spectrometer.

In this study, we can see that the wavelength shifts in the LMR effect were different for each measurement. Figure 4 illustrates the transmission spectra of the sensing devices before and after GZO thin-film coating. The transmitted light intensity of D-shaped fiber decreases after GZO thin film coating. The complex refractive index of GZO thin film can be determined by an ellipsometer. In order to meet the LMR conditions, the MATLAB-based simulation (R2018a, The MathWorks, Inc, Natick, MA, USA) of the refractive index ($n$) and extinction coefficient ($k$) of GZO thin film is performed in the visible wavelength and near infrared (NIR) range. Figure 5 indicates that GZO film meets the conditions of LMR generation, as the refractive index is $n = 2.051$ and the extinction coefficient is $k = 0.0003$, respectively.

Using the LMR method as a sensing mechanism can fabricate different liquid RI sensors. The proposed LMR-type sensor was analyzed by using the wavelength interrogation method. The sensitivity ($S_\lambda$) of the LMR sensor is defined as the following formula:

$$S_\lambda = \frac{\Delta\lambda_{shift}}{\Delta n},$$

(4)

where the $\Delta\lambda_{\text{shift}}$ is the resonance wavelength shift, and $\Delta n$ is the variation in the analyte refractive index.

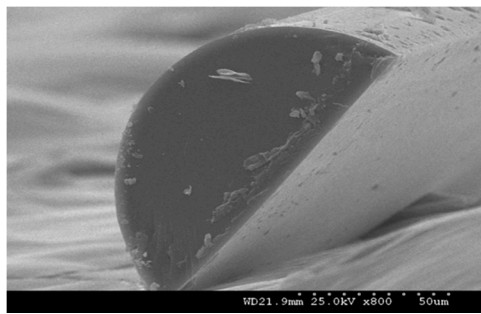

**Figure 2.** SEM image of D-shaped fiber coated with GZO thin film.

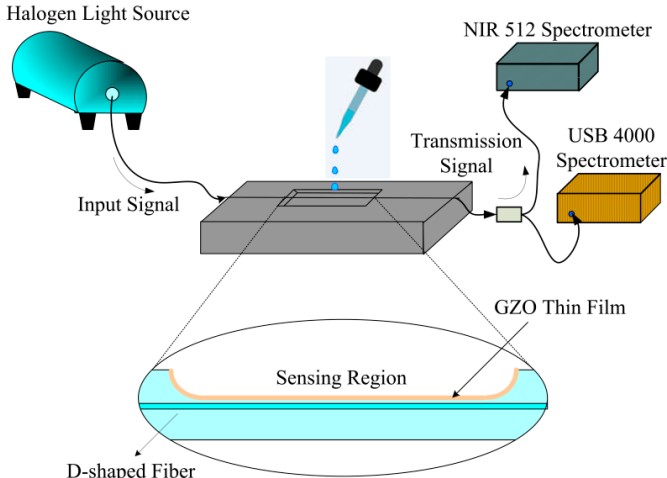

**Figure 3.** Experimental setup for LMR-based refractive index and salinity sensing.

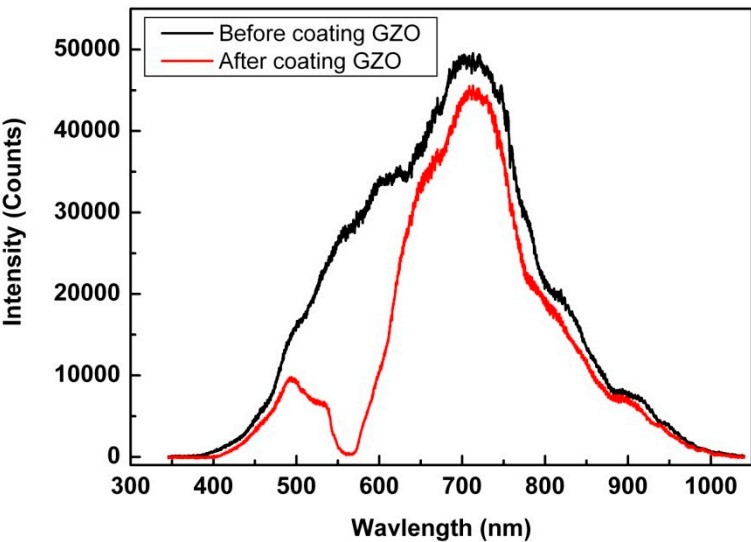

**Figure 4.** Optical transmittance of D-shaped fiber before and after GZO coatings.

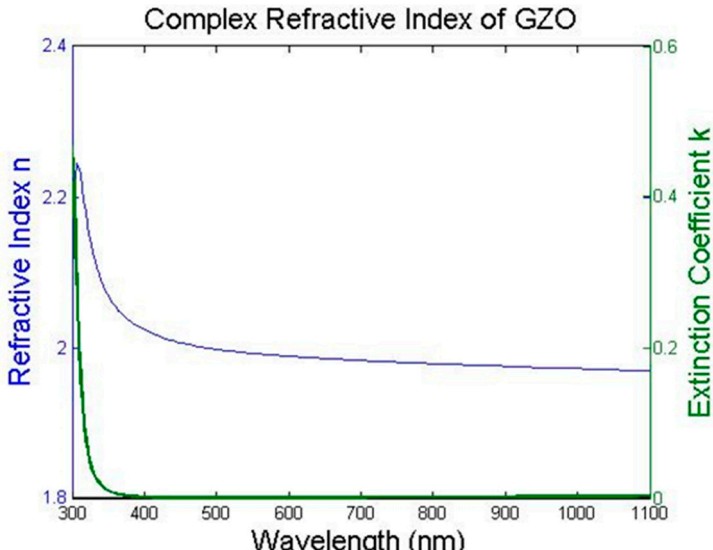

**Figure 5.** Simulation of the refractive index and extinction coefficient of GZO thin film.

When the RI of the surrounding medium changes, the effective refractive index of light propagating in GZO film will vary. For LMR-based RI sensor, the resonance wavelengths shift due to the change of phase-matching conditions. Figure 6 shows the spectral response of a D-shaped fiber sensor with different RI solutions. We changed the RI of the sensing solutions in the sensing section. The proposed fiber-optic RI sensor has an obvious LMR absorption peak when the transmitted spectrum is in the range of 1050–1300 nm. It should be noted that the refractive indices of the sensing solutions were measured using an Abbe refractometer. The range of the refractive indices of the sensing solutions was 1.333 to 1.392. The fiber-optic sensing region was immersed into sensing solutions, and the transmission spectrum data was saved, respectively. The variation in the transmitted optical spectrum of LMR fiber sensor with GZO coating was measured from different concentration NaCl solutions. This result shows that the surrounding refractive index (SRI) is increased, the transmission spectrum of LMR fiber sensor reveals resonance absorption peaks toward longer wavelengths.

As mentioned in the literature [31], the sensitivity of an LMR-based sensor increases with the SRI. The resonance wavelength of the LMR sensor generated with a GZO coating is represented as a function of the RI. Figure 7 depicts the sensitivity of LMR-type RI fiber-optic sensor. A red-shift in the spectrum was observed and the resonance wavelength shift was 214.6 nm, as shown in Figure 7. By taking the curve fitting, the highest sensitivity of the proposed LMR-type sensor was 3637.8 nm/RIU. Here, RIU means a refractive index unit. A resolution of $2 \times 10^{-5}$ RIU was achieved. This work demonstrated an improvement of 25 times in sensitivity with respect to a previous publication by using a D-shaped long period fiber grating (LPFG) sensor [41]. When these results were summarized, we found that GZO coated onto the D-shaped fibers supported the generation of LMR absorption peaks. As the surrounding refractive index sensing solutions changed, the LMR absorption peak shifted toward the longer wavelength side. The results show the sensitivity of the LMR-type fiber sensor is about 4.8 times higher than that of ZnO based LMR sensor [42]. The performance of LMR-type sensors can be obtained using a curve fitting method to determine the sensitivity corresponding to the thickness of GZO films. These results also reveal that the thickness of GZO thin film grew thinner, while the sensitivity of the LMR-type sensors increased. The sensitivity of LMR-type sensor as a function of thin film thickness is shown in Figure 8. Film thickness is an important factor to affect LMR sensor's performance.

The LMR-type RI sensor is able to detect the salinity variations. The experimental setup is the same as discussed in RI sensing. The transmission spectrum of the LMR type fiber-optic sensor is represented as a function of the NaCl liquid salinity, as shown in Figure 9. This device shows a resonance wavelength (LMR wavelength) that has a red-shift for higher surrounding salinity indices

in salinity ranges from 0 to 250%. It can be seen that the total shift in the resonance wavelength is about 194 nm. Villar et al. [31] reported that the variation of the imaginary part (extinction coefficient k) in thin film affects the shape of the LMR curves. The sensitivity is a function of salinity units (SU) as shown in Figure 10. The experimental results show the sensitivity of an LMR-based salinity sensor is 0.964 nm/SU. The correlation coefficient of linearity is high enough to evaluate the degree of salinity. In this study, GZO thin film has been demonstrated as an LMR material for RI and salinity measuring applications. Therefore, the use of sputtering GZO thin film as supporting material for various LMR-type sensors is feasible in the current work.

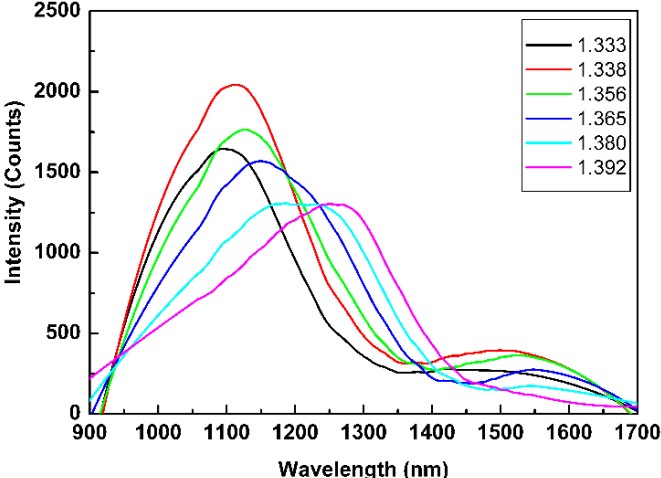

**Figure 6.** Spectrum of LMR type liquid refractive index fiber-optic sensor with different sensing solutions.

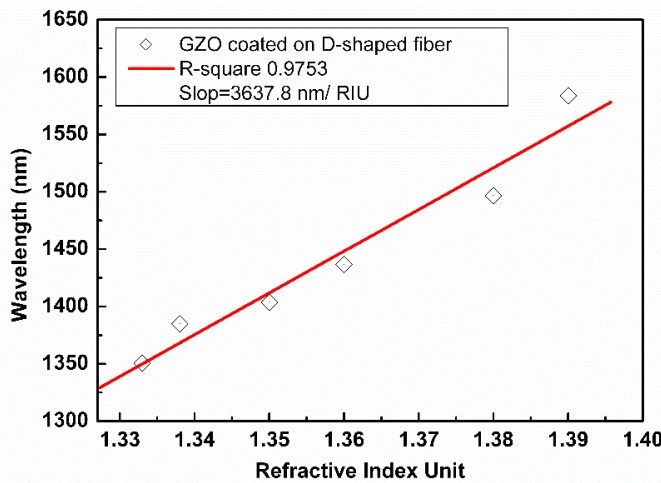

**Figure 7.** The sensitivity of the LMR type liquid refractive index sensor.

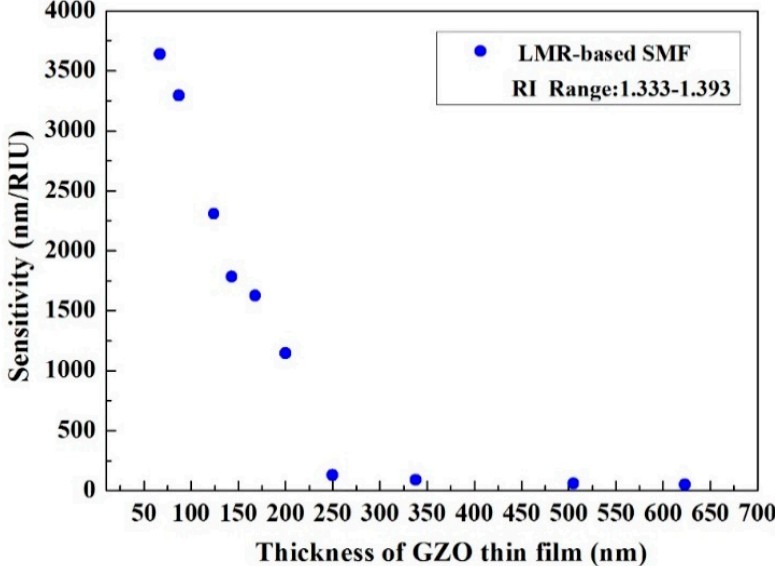

**Figure 8.** The sensitivity of LMR-type sensors as a function of film thickness.

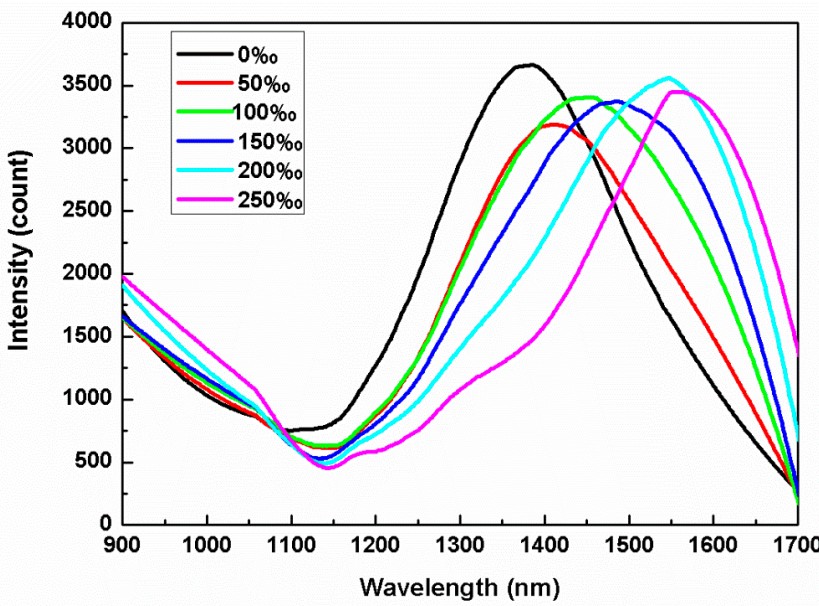

**Figure 9.** The spectra of the LMR type fiber-optic sensor with different salinity solutions.

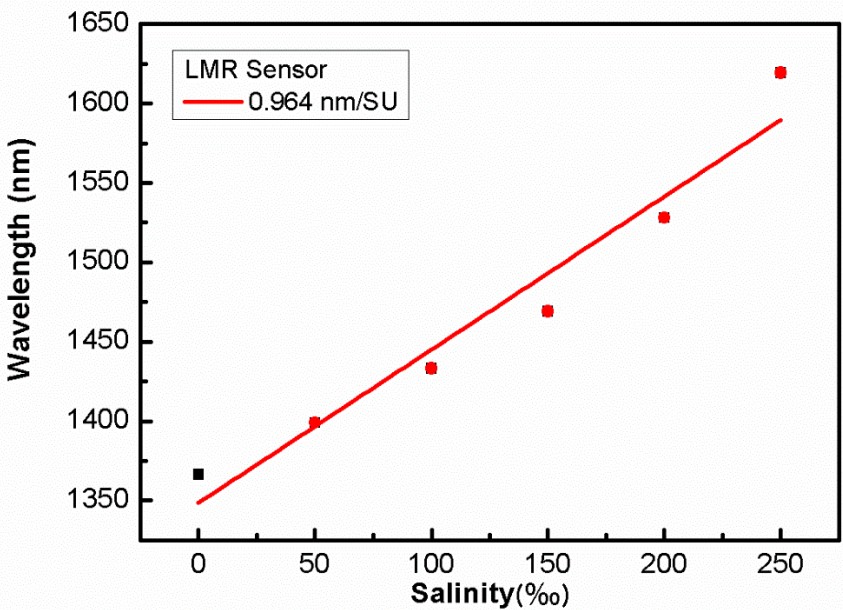

**Figure 10.** The sensitivity of the LMR type salinity sensor.

## 4. Conclusions

We proposed an LMR-type refractive index and salinity sensor by using side-polishing fiber and thin-film coating techniques. GZO thin film is deposited on the polishing surface of D-shaped fibers to produce LMR effect. The optical fiber thickness was 74.7 μm and a GZO film thickness was 67 nm. Thin film coated on the polished surface of the D-shaped fiber can be developed for LMR type liquid refractive index and salinity sensors. The sensitivity of 3637.8 nm/RIU was achieved for an LMR-type RI sensor. We also found that the sensitivity of the proposed salinity sensor for the sensing liquid was 0.964 nm/SU. The proposed LMR type fiber-optic sensor demonstrates a higher sensitivity in the RI and salinity measurements. Furthermore, the proposed sensor design can be readily adapted through modification of the thin-film coatings to detect other chemical species.

**Author Contributions:** Conceptualization and Methodology, C.-L.T.; Validation, C.-Y.L. and T.-C.M.; Formal Analysis, C.-L.T. and C.-Y.L.; Data Curation, C.-L.T., T.-C.M. and C.-Y.L.; Writing—Original Draft Preparation, C.-L.T.; Writing—Review and Editing, C.-L.T. All authors have read and agreed to the published version of the manuscript.

**Funding:** This research was funded by the Ministry of Science and Technology of Taiwan under Contract No. MOST 108-2622-E-035-009-CC3 and Ministry of Education (MOE) Research Project (19M22032).

**Acknowledgments:** The authors are grateful for the Precision Instrument Support Center of Feng Chia University in providing SEM analytical facilities.

**Conflicts of Interest:** The authors declare no conflict of interest.

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
