# Peer review of "Lossy Mode Resonance Sensors Fabricated by RF Magnetron Sputtering GZO Thin Film and D-Shaped Fibers"

_coatings, doi:10.3390/coatings10010029_

Round 1

Reviewer 1 Report

The paper entitled "Lossy Mode Resonance Sensors by RF Magnetron Sputtering GZO Thin Film Coated on D-shaped Fiber" by Tien and Li presents a liquid refractive index (RI) and salinity sensor based on lossy mode resonance (LMR) which combines fiber-optic side-polishing and radio-frequency (RF) sputtering techniques. The proposed LMR type fiber-optic sensor demonstrates a higher sensitivity for the liquid refractive index and salinity measurements. The paper is well-written and well-organized, however, please provide Figures (4-10) at better quality. 

Author Response

Reply to Reviewer 1

The paper entitled "Lossy Mode Resonance Sensors by RF Magnetron Sputtering GZO Thin Film Coated on D-shaped Fiber" by Tien and Li presents a liquid refractive index (RI) and salinity sensor based on lossy mode resonance (LMR) which combines fiber-optic side-polishing and radio-frequency (RF) sputtering techniques. The proposed LMR type fiber-optic sensor demonstrates a higher sensitivity for the liquid refractive index and salinity measurements. The paper is well-written and well-organized, however, please provide Figures (4-10) at better quality.

Response: We thank the reviewer’s valuable comment. Figures 4-10 have been modified from the revised manuscript.

Reviewer 2 Report

Review Report coatings-679352

This paper presents the details of lossy mode resonance sensors fabricated with RF magnetron sputtering of GZO thin film on D-shaped fiber. The idea in novel and interesting and have potential for publication. However, there are serious problems in manuscript structuring and data explanations that need to be seriously considered before publication. Hence, I want to take decision after major revision of the comments provided in the next.

Comments:

The title gives incomplete and vague information about the manuscript. Reconsider the title because it has structural problems. Report some of the most important results in the last paragraph of the introduction section. The text explained in methods should be a part of introduction or literature review sections. Methods should contain only materials and methods used for the research design and experiments. I believe the authors are explaining unnecessary details in the manuscript. I want the authors to be precise and explain the results and experiments analytically. How did the authors fabricate the sensors? How did the authors fabricate GZO films? What techniques were used to characterize GZO films? I believe the authors need to precisely re-write the whole experiment/methods section for clarity. The existing section in not acceptable. Follow the formal MDPI format and instructions to the authors for manuscript writing. The authors do not provide the scientific explanation to the reported facts in the manuscript and jump to the next section. For instance, no evident conclusions have been drawn from Figure 4, so on and so forth. Similarly, there are many Figures but none of the figures are scientifically analyzed in the manuscript. Provide an analytical explanation to each figure with scientific proofs and descriptions. At this moment, the paper is mere a narrations of facts with no scientific soundness. There are multiple methods to fabricate GZO thin films such as hydrothermal, microwave-assisted, molecular beam epitaxy, and atomic layer deposition. The reference papers for the said methods are provided below. I want the authors to cite the following papers in the manuscript and some other papers with different methods of GZO growth and explain why the authors choose RF magnetron sputtering over these methods? Four of the multiple methods are: Rana, A.S and Kim, H.S., 2018. NH4OH Treatment for an optimum morphological trade-off to hydrothermal Ga-doped n-ZnO/p-Si heterostructure characteristics. Materials, 11(1), p.37. Ko, H.J., Chen, Y.F., Hong, S.K., Wenisch, H., Yao, T. and Look, D.C., 2000. Ga-doped ZnO films grown on GaN templates by plasma-assisted molecular-beam epitaxy. Applied Physics Letters, 77(23), pp.3761-3763. Rana, A.S., Shahid, A., Lee, J.Y. and Kim, H.S., 2018. High‐Power Microwave‐Assisted Ga Doping, an Effective Method to Tailor n‐ZnO/p‐Si Heterostructure Optoelectronic Characteristics. physica status solidi (a), 215(5), p.1700763. Park, S.M., Ikegami, T. and Ebihara, K., 2006. Effects of substrate temperature on the properties of Ga-doped ZnO by pulsed laser deposition. Thin Solid Films, 513(1-2), pp.90-94.

Minor Comments

Once an affiliation is defined, always use the abbreviation rather than the acronym. Provide reference to the statement in line 30.

Author Response

Reply to the reviewer 2:

1. The title gives incomplete and vague information about the manuscript. Reconsider the title because it has structural problems.

Response: We thank the reviewer’s valuable comment. We have modified the title as “Lossy Mode Resonance Sensors Fabricated by RF Magnetron Sputtering GZO Thin Film and D-shaped Fibers”.

2. Report some of the most important results in the last paragraph of the introduction section.

Response: We added the most important results as follows:

In this work, we focus on the gallium-doped zinc oxide (GZO) as LMR-supporting thin film material. Zinc oxides have already been widely studied for fabrication of various sensors [29]. If refractive index of thin-film layer is sensitive to the surrounding medium, a wavelength shift can be observed by the LMR effect. The proposed LMR type fiber-optic sensor demonstrates a higher sensitivity for the liquid refractive index and salinity. Therefore, the development of LMR-based RI sensors is helpful for the fabrication of different kinds of sensors.

3. The text explained in methods should be a part of introduction or literature review sections. Methods should contain only materials and methods used for the research design and experiments. I believe the authors are explaining unnecessary details in the manuscript. I want the authors to be precise and explain the results and experiments analytically.

Response: Thanks for your suggestion. We have revised the manuscript and explained the experiment results in more detail. To explain these results, we have also added four references to evidence our experimental results.

4. How did the authors fabricate the sensors? How did the authors fabricate GZO films? What techniques were used to characterize GZO films? I believe the authors need to precisely rewrite the whole experiment/methods section for clarity. The existing section in not acceptable. Follow the formal MDPI format and instructions to the authors for manuscript writing.

Response: Thanks for your suggestions. We have added two subsections in the section of “Materials and Methods“ to describe the experiment of GZO thin-film coatings and the fabrication of the LMR fiber-optic sensors for the liquid refractive index and salinity measurements. In the revised manuscript, we have followed the reviewer’s comments to rewrite the whole experiment and to explain the results in more detail.

5. The authors do not provide the scientific explanation to the reported facts in the manuscript and jump to the next section. For instance, no evident conclusions have been drawn from Figure 4, so on and so forth. Similarly, there are many Figures but none of the figures are scientifically analyzed in the manuscript. Provide an analytical explanation to each figure with scientific proofs and descriptions. At this moment, the paper is mere a narrations of facts with no scientific soundness.

Response: We thank the reviewer’s valuable comment. Figures 4-10 have been modified from the revised manuscript. An analytical explanation to each figure with scientific description has been added and the evidence was cited from the related literature. For this study, GZO thin film has been demonstrated as an LMR material for RI and salinity sensing applications. The use of sputtering GZO thin film as supporting material for various LMR-type fiber-optic sensors is feasible and it exhibits a high sensitivity.

6. There are multiple methods to fabricate GZO thin films such as hydrothermal, microwave assisted, molecular beam epitaxy, and atomic layer deposition. The reference papers for the said methods are provided below. I want the authors to cite the following papers in the manuscript and some other papers with different methods of GZO growth and explain why the authors choose RF magnetron sputtering over these methods? Four of the multiple methods are:

Rana, A.S and Kim, H.S., 2018. NH4OH Treatment for an optimum morphological trade-off to hydrothermal Ga-doped n-ZnO/p-Si heterostructure characteristics. Materials, 11(1), p.37. Ko, H.J., Chen, Y.F., Hong, S.K., Wenisch, H., Yao, T. and Look, D.C., 2000. Ga-doped ZnO films grown on GaN templates by plasma-assisted molecular-beam epitaxy. Applied Physics Letters, 77(23), pp.3761-3763. Applied Physics Letters, 77(23), pp.3761-3763. Rana, A.S., Shahid, A., Lee, J.Y. and Kim, H.S., 2018. High‐Power Microwave‐Assisted Ga Doping, an Effective Method to Tailor n‐ZnO/p‐Si Heterostructure Optoelectronic Characteristics. physica status solidi (a), 215(5), p.1700763. Park, S.M., Ikegami, T. and Ebihara, K., 2006. Effects of substrate temperature on the properties of Ga-doped ZnO by pulsed laser deposition. Thin Solid Films, 513(1-2), pp.90-94.

Response: We thank the reviewer’s valuable comments. There are serval methods to deposit GZO thin films. We have revised the manuscript and added these papers as references. We choose RF magnetron sputtering technique because of offering dense, uniform, and well-adhered films. It has controllable parameters and the most effective processes for the deposition of high quality thin-film materials.

Minor Comments

1) Once an affiliation is defined, always use the abbreviation rather than the acronym.

Response: Thanks for your suggestion. We have corrected the text in the revised manuscript.

2) Provide reference to the statement in line 30.

Response: Thanks for your comment. We have added a reference paper [6].

Round 2

Reviewer 2 Report

I have reviewed the manuscript and I believe that it is ready for publication. Hence, I want to accept the manuscript publication in present form.